# A malaria parasite phospholipase facilitates efficient asexual blood stage egress

Abhinay Ramaprasad[1], Paul-Christian Burda[2,3,4], Konstantinos Koussis[1], James A. Thomas[5], Emma Pietsch[2,3,4], Enrica Calvani[6], Steven A. Howell[7], James I. MacRae[6], Ambrosius P. Snijders[7], Tim-Wolf Gilberger[2,3,4], Michael J. Blackman[1,5]*

1 Malaria Biochemistry Laboratory, The Francis Crick Institute, London, United Kingdom, 2 Centre for Structural Systems Biology, Hamburg, Germany, 3 Bernhard Nocht Institute for Tropical Medicine, Hamburg, Germany, 4 University of Hamburg, Hamburg, Germany, 5 Faculty of Infectious and Tropical Diseases, London School of Hygiene & Tropical Medicine, London, United Kingdom, 6 Metabolomics Science Technology Platform, The Francis Crick Institute, London, United Kingdom, 7 Proteomics Science Technology Platform, The Francis Crick Institute, London, United Kingdom

* mike.blackman@crick.ac.uk

## Abstract

Malaria parasite release (egress) from host red blood cells involves parasite-mediated membrane poration and rupture, thought to involve membrane-lytic effector molecules such as perforin-like proteins and/or phospholipases. With the aim of identifying these effectors, we disrupted the expression of two *Plasmodium falciparum* perforin-like proteins simultaneously and showed that they have no essential roles during blood stage egress. Proteomic profiling of parasite proteins discharged into the parasitophorous vacuole (PV) just prior to egress detected the presence in the PV of a lecithin:cholesterol acyltransferase (LCAT; PF3D7_0629300). Conditional ablation of LCAT resulted in abnormal egress and a reduced replication rate. Lipidomic profiles of LCAT-null parasites showed drastic changes in several phosphatidylserine and acylphosphatidylglycerol species during egress. We thus show that, in addition to its previously demonstrated role in liver stage merozoite egress, LCAT is required to facilitate efficient egress in asexual blood stage malaria parasites.

## Author summary

Malaria kills over half a million people every year worldwide. The disease is caused by a single-celled parasite called *Plasmodium falciparum* that grows and multiplies within a bounding vacuole, inside red blood cells of infected individuals. Following each round of multiplication, the infected cell is ruptured in a process known as egress to release a new generation of parasites. Egress is required for the disease to progress and is orchestrated by the parasite, which sends out various molecules to puncture and destroy the membranes of the vacuole and the red blood cell. However, little is known about these molecules. In this work, we set out to identify these molecules by using genetic and proteomics approaches. After first showing that two putative pore-forming parasite proteins are not required for egress, we next screened the molecules the parasite releases during egress and identified a parasite enzyme called LCAT present in the vacuole. We found that mutant

**Data Availability Statement:** All relevant data are within the manuscript and its Supporting Information files. All statistical analysis is available

as R code in https://github.com/a2g1n/LCATxcute. Metabolomics data have been deposited to the EMBL-EBI MetaboLights database (DOI: 10.1093/nar/gkz1019, PMID:31691833) with the identifier MTBLS8011 (https://www.ebi.ac.uk/metabolights/MTBLS8011).

**Funding:** AR was funded by a Marie Skłodowska Curie Individual Fellowship (Project number 751865). The work was also supported by funding to MJB from the Wellcome Trust (220318/A/20/Z) and the Francis Crick Institute (https://www.crick.ac.uk/) which receives its core funding from Cancer Research UK (CC2129), the UK Medical Research Council (CC2129), and the Wellcome Trust (CC2129). For the purpose of Open Access, the author has applied a CC BY public copyright licence to any Author Accepted Manuscript version arising from this submission. The work was further supported by Wellcome ISSF2 funding to the London School of Hygiene and Tropical Medicine. PCB acknowledges funding by the German Research Foundation (DFG project number 414222880), whilst PB and TWG acknowledge the support of the DFG research networks SPP 2225. The funders had no role in study design, data collection and analysis, decision to publish, or preparation of the manuscript.

**Competing interests:** The authors have declared that no competing interests exist.

parasites that were unable to make LCAT clumped together and could not escape the infected cell properly. As a result, we saw a reduction in the rate at which these parasites spread through the red blood cells. Taken together, our findings suggest that *P. falciparum* needs LCAT to efficiently escape from red blood cells.

## Introduction

During the asexual blood stages (ABS) of their life cycle, malaria parasites grow and replicate asexually within a parasitophorous vacuole (PV) in host red blood cells (RBCs). At the end of each cycle of intraerythrocytic development, invasive merozoites are released from the host cell in a coordinated lytic process known as egress, to invade fresh RBCs. Egress involves a rapid sequence of events resulting in the rupture of the two bounding membranes, the PV membrane (PVM) and the RBC membrane (RBCM). Minutes before egress, the PV rounds up without swelling in a calcium-dependent process [1], followed by PVM leakage and rupture, then subsequent poration and rupture of the RBCM [1–6]. Egress is initiated by activation of a cGMP-dependent protein kinase (PKG) [7] in coordination with a calcium-dependent protein kinase called CDPK5 [8] by triggering the discharge of specialised parasite secretory organelles called micronemes and exonemes. A subtilisin-like parasite protease (SUB1) is released from the exonemes into the PV lumen where it initiates a proteolytic cascade by cleaving and activating several effector molecules including members of the papain-like SERA protein family and several components of the merozoite surface [9–13]. Activated SERA6 precisely cleaves β-spectrin in the RBC cytoskeleton to bring about the final step of RBCM rupture [14]. Despite these many insights, the molecular basis for the events that precede RBCM rupture, them being PVM rupture and RBCM poration, remains unknown. Membrane-lytic effector molecules such as perforin-like proteins (PLPs) and phospholipases likely bring about these membrane-degradative events. Previous evidence [15] showing that RBCM poration occurs immediately following PVM rupture has led us to postulate that the same effector molecules may mediate both events. These effectors may either be released from secretory organelles just prior to egress or are constitutively resident within the PV waiting to be activated by the SUB1-initiated proteolytic cascade.

Pore-forming proteins of the membrane attack complex component/perforin (MACPF) superfamily disrupt membranes by inserting their characteristic MACPF domain into the target phospholipid bilayer to form a transmembrane channel [16]. Five *Plasmodium* proteins possessing MACPF domains, annotated as perforin-like proteins (PPLPs), are expressed significantly in the sexual gametocyte stages that transmit the parasite to the mosquito vector, as well as in the mosquito ookinete and sporozoite stages; however, the PPLPs are transcribed only at low levels in ABS (PlasmoDB v46, [17]). Consistent with this, in previous genetic analyses in which each of the five PPLP genes was individually disrupted, the genes were shown to be dispensable for ABS growth but were instead implicated in other parasite life cycle transitions [18]. PPLP1 is required for sporozoite egress from transient vacuoles formed by the parasite in host hepatocytes [19–21], while PPLP2 is required for gametocyte egress [22,23]. Despite these findings, other studies have shown that PPLP1 and PPLP2 are detectable in ABS parasites [23,24] and biochemical and small molecule inhibitor evidence suggested a possible role for both PPLPs in membrane poration during merozoite egress [24,25]. One plausible explanation for this apparent discrepancy is that PPLP1 and PPLP2 perform redundant functions in ABS, in which case disruption of any one gene might be functionally complemented by the other. However, this possibility has not been examined.

Phospholipases (PLs) mediate membrane lysis through hydrolytic cleavage of either of the acyl chains (phospholipases A1, A2 and B) or phosphodiester bonds in the glycerol backbone (phospholipases C and D) of membrane phospholipids. Phospholipases often aid exit of intracellular bacteria from the cellular vacuole [26], and can also cleave fatty acids from phospholipids to alter membrane curvature through localised phospholipid asymmetry [27], raising the possibility of similar roles during the extensive morphological changes associated with malarial egress. A systematic functional analysis of 20 out of 27 putative phospholipases in the human malaria parasite, *Plasmodium falciparum* suggested a high degree of functional redundancy, with only five being found to possibly be essential in ABS [28]. Of those phospholipases deemed dispensable in ABS, a secreted phospholipase with a lecithin:cholesterol acyltransferase (LCAT)-like domain has previously been shown to disrupt membranes during mosquito and liver stages of the parasite in the rodent malarial species *P. berghei* [29,30]. LCAT was found to be expressed on the surface of sporozoites, and LCAT-null sporozoites had a reduced capacity to egress from oocysts and migrate through host hepatocytes. LCAT localises to the PV and PVM following invasion of hepatocytes and LCAT-null parasites were defective in liver stage schizont egress due to impaired PVM rupture that prevents or delays merozoite release. Whether LCAT plays any role in *Plasmodium* blood stage egress remains unclear.

Here, we first rule out any essential requirement for PPLP1 and PPLP2 in ABS egress. We then profile the repertoire of proteins that are discharged from the micronemes and exonemes at egress to examine whether they include any previously unidentified effector molecule(s) potentially involved in PVM rupture and RBC poration. Our resulting work allows us to demonstrate that LCAT plays a previously unrecognised role in facilitating efficient egress.

## Results

### Perforin-like proteins are dispensable for ABS egress

While genetic ablation of PPLP1 and PPLP2 individually has previously been shown to have no effect on parasite proliferation [20,23], the question remained as to whether the proteins could function in a compensatory manner to mediate membrane poration during egress. To examine this, we attempted to generate a parasite line in which both PPLP1 (PF3D7_0408700) and PPLP2 (PF3D7_1216700) were simultaneously disrupted. To do this, we set about floxing the MACPF-encoding domains of both genes in the DiCre-expressing *P. falciparum* line B11 (Figs 1A and S1A), so that rapamycin (RAP)-mediated activation of the DiCre could be used to simultaneously excise functionally critical regions of both genes. Due to an error in repair plasmid design for *pplp1*, we inadvertently introduced a frameshift mutation and disrupted the gene when floxing its MACPF domain. The resultant PPLP1-null *P. falciparum* line (D1) displayed normal growth (S1B Fig) as expected, and so was used as the background for subsequent floxing of *pplp2*. RAP treatment of the final modified parasite line (called PPLP1:loxNint/PPLP2:loxPint) led to efficient excision of both genetic loci within a single erythrocytic cycle (Fig 1B). As shown in (Fig 1C and 1D), the resulting PPLP1/PPLP2-null mutant parasites displayed normal proliferation rates with no discernible effect on parasite development. To assess RBCM poration, we performed simultaneous time-lapse imaging of both RAP- and mock-treated (stained blue with DAPI) PPLP1:loxNint/PPLP2:loxPint parasites in the presence of fluorescent phalloidin (green), a cyclic peptide that gains access to and binds to the short F-actin filaments within the RBC cytoskeleton upon RBCM poration. These experiments showed that the PPLP1/PPLP2-null schizonts underwent typical RBC poration just prior to egress (S1 Movie). It was concluded unambiguously that neither PPLP1 nor PPLP2 have an essential role in membrane poration or RBC rupture during ABS egress.

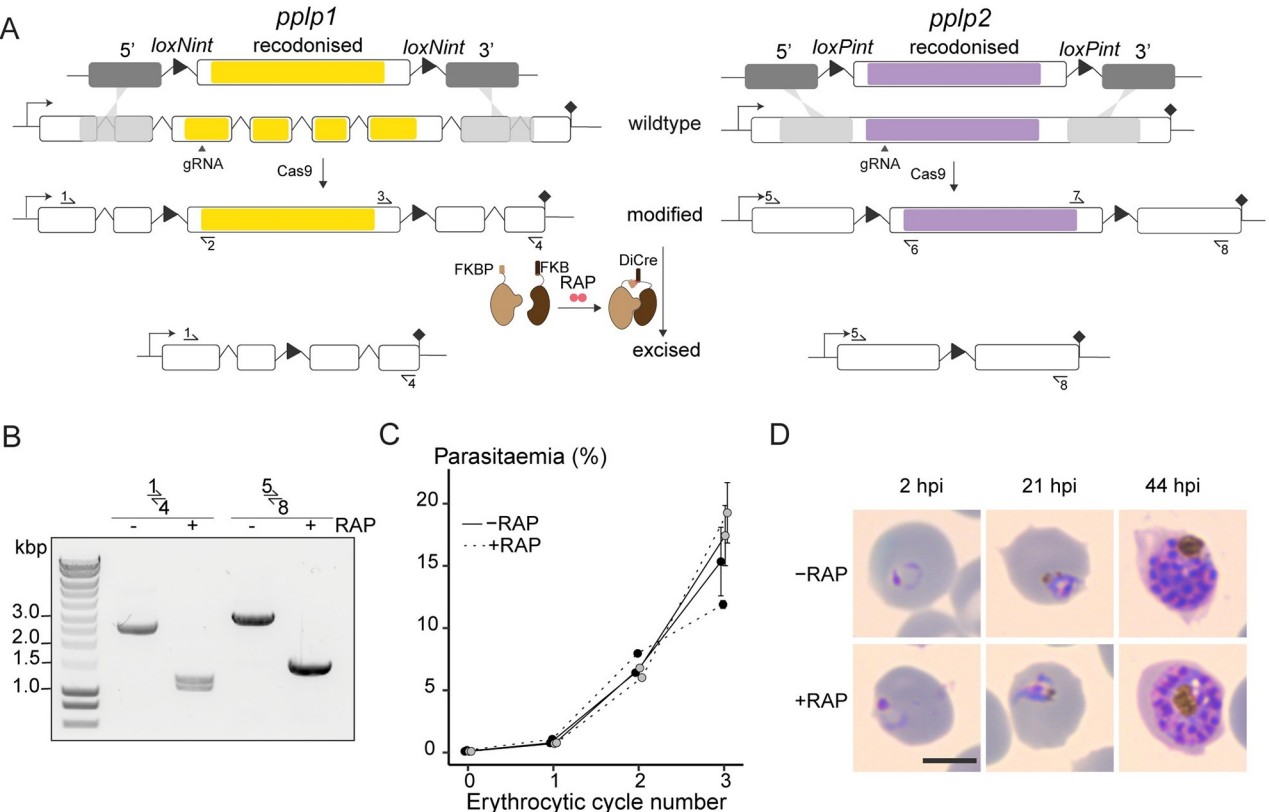

**Fig 1.** *P. falciparum* **PPLP1 and PPLP2 are dispensable for asexual blood stage egress. A)** Strategy used for simultaneous conditional disruption of both PPLP1 and PPLP2 in the parasite line PPLP1:loxNint/PPLP2:loxPint. The MACPF domains of both PLPs (yellow and purple) are floxed by introducing loxN- and loxP- containing introns (loxPints) respectively. Site of targeted Cas9-mediated double-stranded DNA break (marked "gRNA"), left and right homology arms for homology-directed repair (5' and 3') and diagnostic PCR primers (half-arrows 1 to 8) are indicated. RAP-induced dimerization of N- and C- terminal subunits of the Cre enzyme enables Cre-mediated excision of the floxed regions rendering both the genes non-functional. **B)** Diagnostic PCR using primers 1–4 and 5–8 (representative of 3 independent experiments) confirms efficient excision at both loci sampled at 12 h post-RAP (+RAP) or -mock treatment (-RAP) of ring stages. **C)** Replication of mock- (solid line) and RAP-treated (dashed line) parasites from two clonal lines of PPLP1:loxNint/PPLP2:loxPint, C1 and C2, over three erythrocytic cycles (error bars, ± SD, triplicate RAP treatments with different blood sources). There is no significant difference in replication rates. **D)** Light microscopic images of Giemsa-stained PPLP1:loxNint/PPLP2:loxPint parasites following mock- or RAP-treatment at ring stage in cycle 0 (representative of 2 independent experiments). PPLP1/PPLP2-null parasites exhibit normal parasite development. Scale bar, 5 μm.

## Differential proteomics of SUB1-null schizonts identifies proteins discharged into the PV at egress

Minutes before merozoite egress, malaria parasites discharge the contents of micronemes and exonemes onto the merozoite surface or into the PV lumen, where some of the discharged proteins perform specific tasks to bring about egress. Discharge of the exoneme protease SUB1 is required to mediate both PVM and RBCM rupture [14]. As a consequence, SUB1-null parasites arrest in a state in which microneme and exoneme discharge has occurred normally, but the PVM and RBCM remain intact, resulting in the discharged organelle contents remaining trapped within an intact PV. This is in contrast to treatment of schizonts with the PKG inhibitor 4-[7-[(dimethylamino)methyl]-2-(4-fluorphenyl)imidazo[1,2-α]pyridine-3-yl]pyrimidin-2 amine (compound 2, C2); this leads to a similar morphological phenotype, with merozoites trapped within an intact PVM and RBCM, but in this case microneme/exoneme discharge is blocked (Fig 2A). We reasoned that these contrasting phenotypes provided an opportunity to

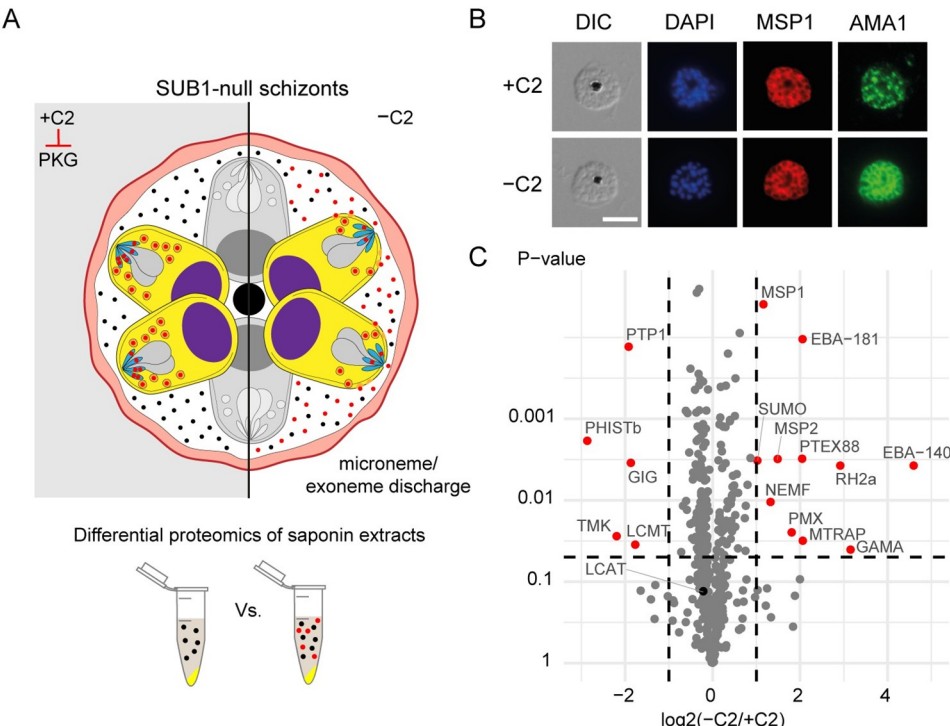

**Fig 2. Proteomic identification of organelle proteins discharged during egress. A)** Schematic of experimental design to identify components of micronemes and exonemes released prior to egress. SUB1-null schizonts were incubated in C2 (+C2) to prevent discharge of microneme and exoneme proteins (marked in red). Upon washing away C2 (-C2), micronemes and exonemes release their contents into the PV and are retained there in SUB1-null schizonts. Comparisons of the proteomic profiles of saponin extracts of the C2-arrested and C2-washed SUB1-null schizonts is expected to show differences in the abundance of microneme and exoneme proteins. **B)** Immunofluorescence assay showing AMA1 localisation (green). In the presence of C2, AMA1 is restricted to the micronemes at the apical ends of the merozoites. In contrast, upon C2 removal, AMA1 translocates onto the merozoite cell surface. Parasite surface marker MSP1 (red), DAPI-stained nuclei (blue) and DIC, differential interference contrast are also shown. Scale bar, 5 μm. **C)** Volcano plot showing enrichment of 11 proteins (red) in the C2-washed schizonts compared to C2-arrested SUB1-null schizonts (values averaged from biological triplicates).

selectively identify the repertoire of microneme and exoneme proteins that are discharged into the PV at egress. Of particular interest, we reasoned that this group of discharged proteins might include amongst them previously unidentified effector molecule(s) involved in PVM rupture and RBC poration.

To selectively identify these 'trapped' components, we allowed SUB1-null schizonts (created by inducing ablation of SUB1 expression in the previously-reported inducible knockout line, SUB1HA3:loxP [14]) to mature in the presence of C2, then washed away the C2 for 20 minutes to allow organelle discharge before finally releasing the PV contents using saponin lysis, which disrupts both the RBCM and PVM but not the parasite plasma membrane. Successful microneme discharge upon washing away C2 was confirmed by observing relocalisation of the micronemal protein AMA1 to the periphery of merozoites (Fig 2B). We then compared the proteomic profiles of the saponin lysates of C2-washed (-C2) and C2-arrested (+C2) parasites to identify those exonemal/micronemal proteins discharged into the PV within the 20-minute window.

A total of 503 parasite proteins were identified, including several established constitutively PV-resident proteins such as the SERA family of papain-like proteins [11,31] and protein

phosphatase UIS2 [32], as well as proteins exported to the RBC cytosol including secreted rhoptry proteins and the PHIST family of exported proteins (S2 Table). The remaining proteins were known parasite cytosolic or nuclear proteins including ribosomal and proteasomal proteins, probably representing contaminants from the parasite fraction. Levels of all these proteins were largely equivalent between the +C2 and -C2 schizonts, as expected for constitutively-expressed PV proteins. A total of only 11 proteins were found to be significantly more abundant in -C2 saponin extracts (p<0.05, more than 2-fold change; see Fig 2B). Apart from a few surface (MSP1 and MSP2), rhoptry neck (RH2a) and PV (PTEX88) proteins, these included the exoneme protease plasmepsin X (PMX) [33,34] which was enriched 4-fold, as well as four membrane proteins that are discharged from micronemes during schizont rupture: EBA-140, EBA-181, GAMA and MTRAP (4-fold to 24-fold enrichment). The enrichment of these bona fide microneme/exoneme proteins in the -C2 samples indicated that our strategy was successful, but no candidates with established membranolytic activity were identified amongst the enriched (organelle-derived) population. However, a lecithin:cholesterol acyltransferase (LCAT) previously implicated in liver stage merozoite egress in the rodent malaria parasite *P. berghei* [30], was detected in all the samples (S2 Table) indicating that this enzyme is constitutively resident in the PV. We chose this protein for further investigation.

## Conditional genetic ablation of LCAT reduces blood stage proliferation

Previous transcriptomic analyses indicate that peak expression of LCAT (PF3D7_0629300) during ABS occurs during schizont development [35]. To investigate the localisation of LCAT in ABS *P. falciparum* parasites, we appended a C-terminal GFP-tag to the endogenous gene using the selection-linked integration (SLI) system [36] (Figs 3 and S2A and S2B). Live fluorescence microscopy of LCAT:GFP schizonts revealed that the fluorescent signal localised to focal structures within the parasite and around developing merozoites, pointing towards a localisation to secretory organelles and the PV (Fig 3). A similar localisation was observed by immunofluorescence analysis (IFA) of fixed parasites expressing LCAT fused to spaghetti monster-Myc (smMyc) tag [37] (S2C, S2D and S2E Fig). Co-localisation analysis with various secretory organellar markers showed that the intracellular LCAT signal was not from rhoptries, micronemes or exonemes (S2F Fig). On the other hand, western blot analysis using rabbit polyclonal

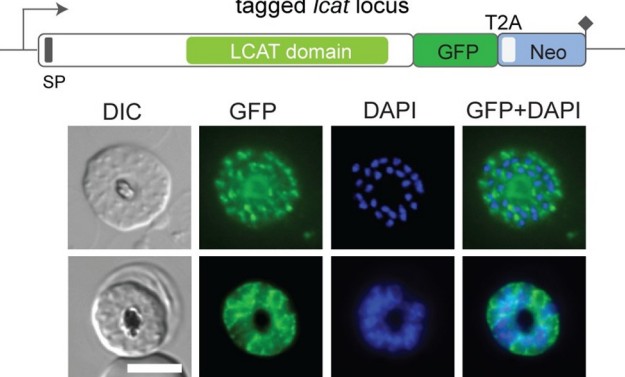

**Fig 3. LCAT localises to secretory vesicles and the PV in blood stage schizonts.** Live-cell microscopy of mature schizonts expressing endogenously tagged LCAT:GFP (green). The GFP signal was detected in focal structures within the parasite (top row) and around developing merozoites (bottom row). Nuclei were stained with DAPI (blue). DIC, differential interference contrast. Scale bar, 5 μm.

antibodies raised against LCAT showed that most of the protein was released to the supernatant upon saponin lysis of schizonts, consistent with a localisation to the PV (S2G Fig).

To conditionally ablate the *lcat* gene, we designed a DiCre-mediated gene disruption strategy that introduces a translational frameshift, truncating the gene to render it non-functional (Figs 4A and S3). For this, we floxed a short 200 bp segment upstream of the putative catalytic domain in the *lcat* gene (PF3D7_0629300) by introducing two closely-opposed loxPint modules, producing a modified inducible LCAT knockout line called LCAT:2loxPint. DiCre-mediated excision of the floxed region was predicted to result in a frameshift mutation that introduces multiple stop codons in the downstream sequence encoding the LCAT catalytic domain. RAP-treatment of two clonal LCAT:2loxPint lines (B10 and F10) resulted in efficient

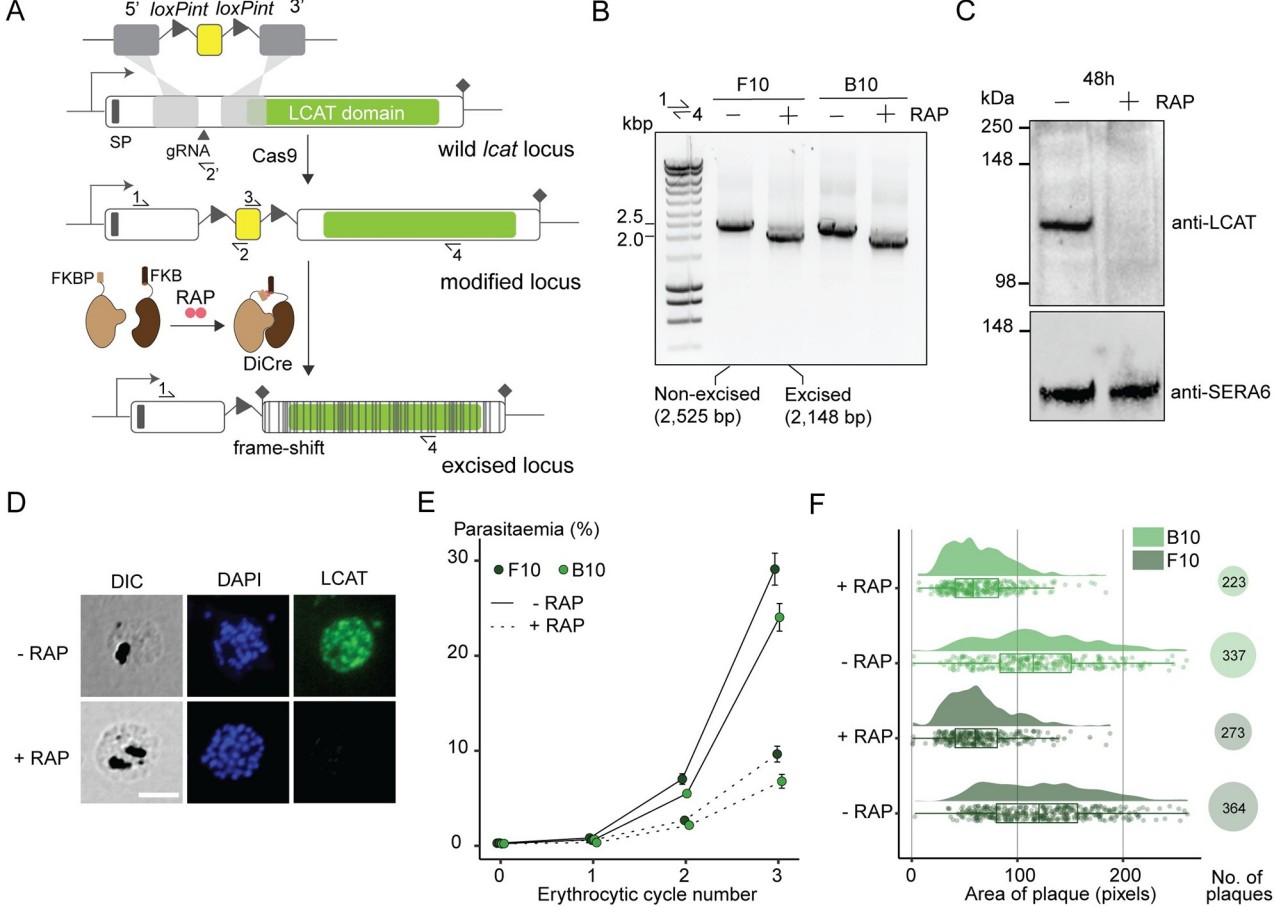

**Fig 4. Genetic ablation of LCAT expression reduces blood stage proliferation. A)** Strategy used for conditional disruption of LCAT in parasite line LCAT:2loxPint. A 200 bp region upstream of the predicted LCAT domain (green) is floxed by introducing two loxPint modules. Predicted secretory signal peptide (SP), site of targeted Cas9-mediated double-stranded DNA break (marked "gRNA"), left and right homology arms for homology-directed repair (5' and 3'), recodonized sequence (yellow) and diagnostic PCR primers (half arrows 1–4) are indicated. RAP-induced DiCre-mediated excision results in frameshift that renders the gene non-functional (grey lines). **B)** Diagnostic PCR 12 h following mock- or RAP-treatment of ring-stage LCAT:2loxPint (representative of 3 independent experiments) confirms efficient gene excision. Expected amplicon sizes are indicated. **C)** Western blots (representative of 3 independent experiments) showing successful RAP-induced ablation of LCAT expression in LCAT:2loxPint parasites sampled at 48 h post invasion. SERA6 was probed as loading control. **D)** IFA of RAP-treated (+RAP) and mock-treated (-RAP) mature LCAT:2loxPint schizonts showing that expression of LCAT is lost following RAP treatment. Scale bar, 5 μm. **E)** RAP-treatment results in reduced replication rate in two clonal lines, F10 and B10, of LCAT:2loxPint parasites. Data shown are averages from triplicate biological replicates using different blood sources (error bars, ± SD). **F)** RAP-treatment results in reduction in both number and area of clonal plaques formed over five erythrocytic cycles (10 days of growth) in LCAT:2loxPint clonal lines (individual points represent the area of each plaque, density plot shows distribution of these points and the boxplot provides median summary statistics).

excision of the floxed sequence (Fig 4B) and loss of LCAT expression as confirmed by western blotting and IFA (Fig 4C and 4D).

The RAP-treated LCAT-null clonal lines displayed a reduced proliferation rate (~66% reduction over 3 erythrocytic cycles) compared to mock-treated controls (Fig 4E). Longer-term viability of the LCAT-null parasites over ~5 erythrocytic cycles as assessed by plaque assay [38] reflected this, with a 25–34% reduction in the number of plaques formed following RAP-treatment, as well as a significant reduction in the average area of these plaques (Fig 4F). It was concluded that LCAT is important for ABS parasite replication *in vitro*.

## Loss of LCAT causes inefficient egress

To explore in more detail the growth defect in LCAT-null parasites, we monitored their development over the course of a single erythrocytic cycle. As shown in Fig 5A, this revealed no obvious impact on intracellular growth or morphology that might explain the proliferation defect (Fig 5A). We then visualised the behaviour of RAP- (stained with Hoechst stain) and mock-treated LCAT:2loxPint schizonts as they underwent egress using time-lapse video microscopy. The mock-treated schizonts displayed the typical morphological changes associated with egress, including PVM swelling and rounding up, followed by PVM rupture within seconds, and finally RBCM rupture and merozoite release. In contrast, LCAT-null schizonts showed clear delays between these sequential events, with reduced merozoite release upon RBCM rupture, the remaining merozoites often appearing clumped together (Fig 5B–D and S2 Movie).

To determine whether the egress phenotype contributes to the lower replication rates in the LCAT-null parasites, fold changes in parasitaemia during egress and invasion were compared in RAP- and mock-treated cultures (Fig 5E). In standard static cultures, RAP-treated parasites showed a two-fold reduction in parasitaemia increase compared to control cultures. This defect was rescued when the experiment was performed under shaking conditions, suggesting that the clumped LCAT-null merozoites can be released and dispersed efficiently under conditions of shear stress. These results also indicated that LCAT-null parasites do not have an intrinsic invasion defect. Taken together, our results show that LCAT-null parasites display a reduced replication rate that can be primarily ascribed to inefficient egress from the host RBC.

To determine whether the abnormal egress phenotype of LCAT-null parasites was due to a defect in membrane poration, we visualized RBCM poration using fluorescent phalloidin in the additional presence of the cysteine protease inhibitor E64 which blocks RBCM rupture but allows PVM rupture and RBCM poration, thus facilitating visualization of poration [2,14]. These experiments showed that whilst RBCM poration did take place in LCAT-null parasites (stained with Hoechst), we observed a small reduction in the proportion that had undergone poration, suggestive of a slightly slower rate of poration in the mutants (Fig 5F and 5G and S3 Movie). We confirmed this phenotype by quantifying porated parasitised RBCs in larger populations using flow cytometry (Fig 5H). Nevertheless, the defect in poration is evidently subtle and is unlikely to explain the high rate of abnormal egress we observe in the mutants. We concluded that LCAT is required for efficient parasite egress in a manner unlikely to be related to RBCM poration.

## Phosphatidylserine and acylphosphatidylglycerol levels change during egress of LCAT-null parasites

Given the predicted role of LCAT as a catalytically active phospholipase, we reasoned that the LCAT-null phenotype likely resulted from a defect in parasite phospholipid modification prior to and/or during egress. To seek insights into this, we studied the phospholipid composition of

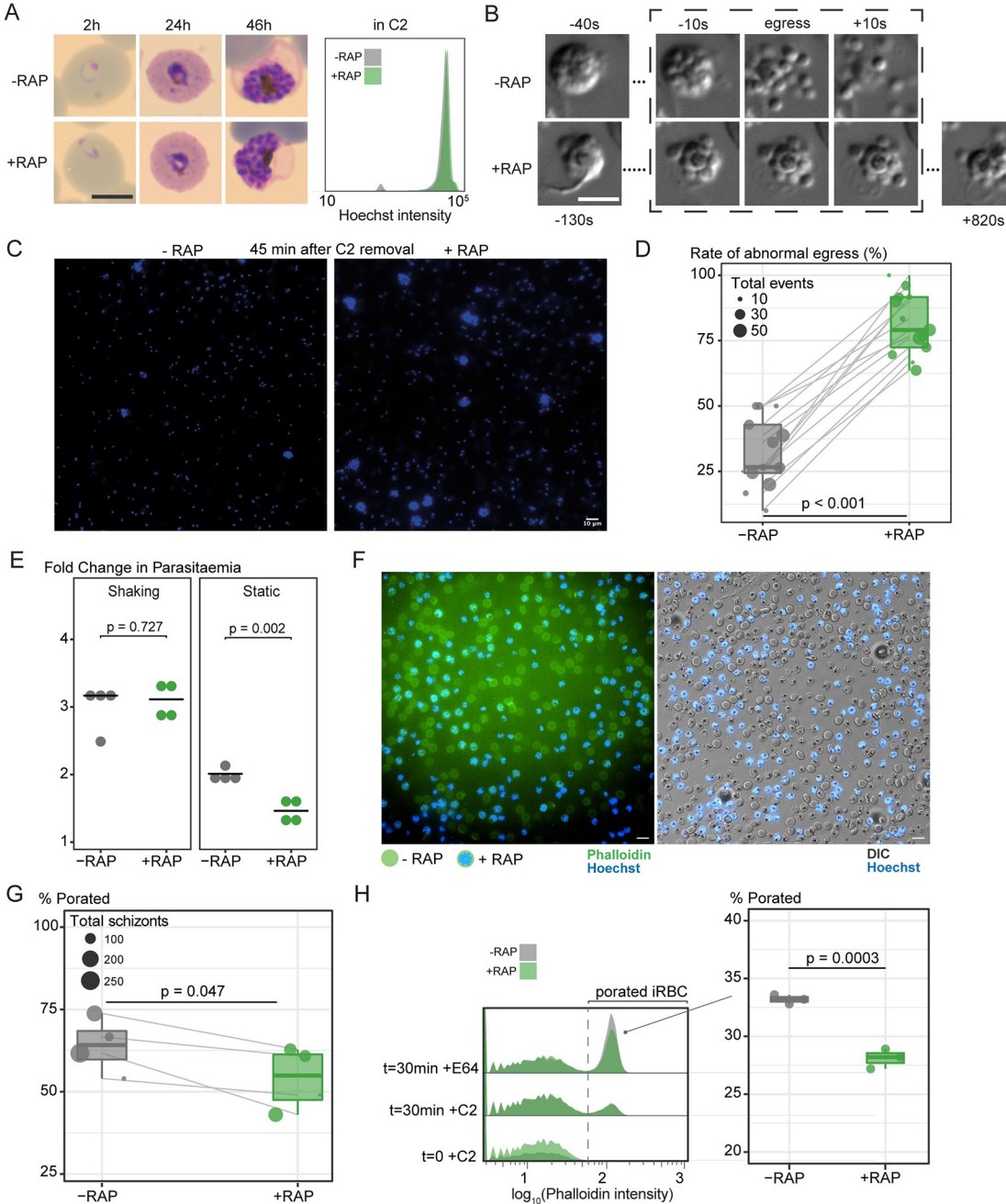

**Fig 5. LCAT is required for efficient asexual blood stage egress. A)** Light microscopic images of Giemsa-stained LCAT:2loxPint parasites following mock- or RAP-treatment at ring stages (representative of 2 independent experiments). LCAT-null parasites exhibit normal parasite development. Inset shows confirmation of normal growth by measuring DNA content of egress-arrested schizonts using flow cytometry. Scale bar, 5 μm. **B)** LCAT-null parasites (+RAP) show delayed onset of egress compared to mock-treated controls and inefficient dispersal of merozoites that resulted in clumped merozoites. This was defined as an abnormal egress event. Scale bar, 5 μm. **C)** DAPI-staining of egressed RAP-treated (+RAP) and mock-treated (-RAP) mature LCAT:2loxPint parasites show persistence of clumped merozoites that are products of abnormal egress in LCAT-null parasites. Scale bar, 10 μm. **D)** LCAT-null schizonts show a higher number of abnormal egress events (as described in Fig 5B) compared to mock-treated schizonts (paired Student's t-test). Each paired datapoint represents a 30–40 min video of RAP- and mock-treated LCAT:2loxPint schizonts (one group randomly stained with Hoechst DNA stain in each video) undergoing egress (from a total of 7 independent experiments). Size of each datapoint represents the total number of egress events (abnormal + normal) counted in the video. **E)** Fold change in parasitaemia after 4 h invasion of mock- (-RAP) and RAP-treated (+RAP) LCAT:2loxPint schizonts under shaking and static conditions. Static cultures show a significantly lower fold change in parasitaemia in RAP-treated parasites compared to mock-treated controls, while show no significant difference between the groups was observed in shaking cultures (error bars, ± SD, four

replicate RAP treatments with different blood sources; individual points represent each replicate). F) Poration of the RBCM occurred in both mock- (RAP-) and RAP-treated (+RAP; stained blue with Hoechst DNA stain) LCAT:2loxPint schizonts visualized as accumulation of a fluorescent phalloidin (green) signal around the RBC circumference, maintained intact by the presence of E64 which inhibits the final step of RBCM rupture. Scale bar, 10 μm. **G)** A subtle decrease in the rate of RBCM poration (paired Student's t-test) was observed in RAP-treated schizonts compared to mock-treated schizonts. Each paired datapoint represents a 30–40 min video of RAP- and mock-treated LCAT:2loxPint schizonts (one group randomly stained with Hoechst DNA stain in each video) undergoing egress in the presence of E64 (from a total of 3 independent experiments). Size of each datapoint represents the total number of schizonts counted in the video. **H)** Flow cytometry analysis showed emergence of porated parasitised RBCs (iRBCs) that emit higher fluorescence intensity from phalloidin following 30 min of egress in the presence of E64 (30 min + E64) after washing off C2. Samples harvested before washing off C2 (0 min + C2) or maintained in C2 for the same 30 min period (30 min + C2) served as controls. A slight but consistent decrease in proportion of porated (iRBCs) was observed in RAP-treated compared to mock-treated schizonts.

LCAT-null mutants, comparing lipid profiles of RAP- and mock-treated LCAT:2loxPint schizonts just prior to egress.

Analysis by quantitative liquid chromatography-coupled mass spectrometry (LC-MS/MS) detected a total of 111 lipid species and found the lipid profiles of both schizont samples to be remarkably similar (S4A Fig). This indicates that loss of LCAT does not have any detectable impact on the phospholipid composition during parasite development up to mature schizont stage.

Next, we profiled the phospholipid content of these schizonts before and immediately following egress to examine phospholipid-level changes occurring during egress (Fig 6A). For this, we first extracted lipids from synchronous, C2-arrested schizont populations of both RAP- and mock-treated parasites (time point before egress, BE). We then released the egress arrest by washing away the C2 and allowed the schizonts to undergo egress for 45 minutes in a small volume of culture media. The entire sample, confirmed by microscopy (Fig 6B) to comprise predominantly free merozoites, ruptured RBC and PV membranes, and the few residual schizonts that did not undergo egress, was subjected to lipid extraction (time point after egress, AE). Pairwise comparisons between the BE and AE samples within LCAT-null and wild type control parasites showed significant changes in abundance of several species belonging to three lipid classes- phosphatidylserine (PS), phosphatidylethanolamine (PE) and acylphosphatidylglycerol (acylPG) (Figs 6C and S4B). Notably, 9 out of 11 PS species detected were significantly enriched (1.5–2 fold) upon egress of the LCAT-null parasites but not during egress of wild type controls (Figs 6C and S5A). Similar enrichment was observed in some PE species in LCAT-null parasites while in contrast wild type egress produced a significant decrease in several PE species. We also observed a striking decrease in all acylPG species (1.5 to 2-fold change) during egress of both LCAT-null and wild type schizonts which suggests that this decrease is normally associated with egress and is independent of LCAT activity (Figs 6C and S5B).

## Discussion

In the minutes leading to ABS parasite egress, the PV is the site of intense proteolytic and membranolytic activity. Secretory vesicles are discharged, the PV rounds up, and the PVM and RBCM are ruptured and porated respectively before final RBCM rupture. PVM rupture and RBCM poration are brought about by unidentified effector molecules that act within a very short time frame, and we speculate that both events may be mediated by the same effector molecule(s). To gain a better understanding of these events during egress, we here studied i) the role of two perforin-like proteins (PPLP1 and PPLP2); ii) a PV-resident phospholipase (LCAT); and iii) changes in protein and phospholipid content that occur during egress.

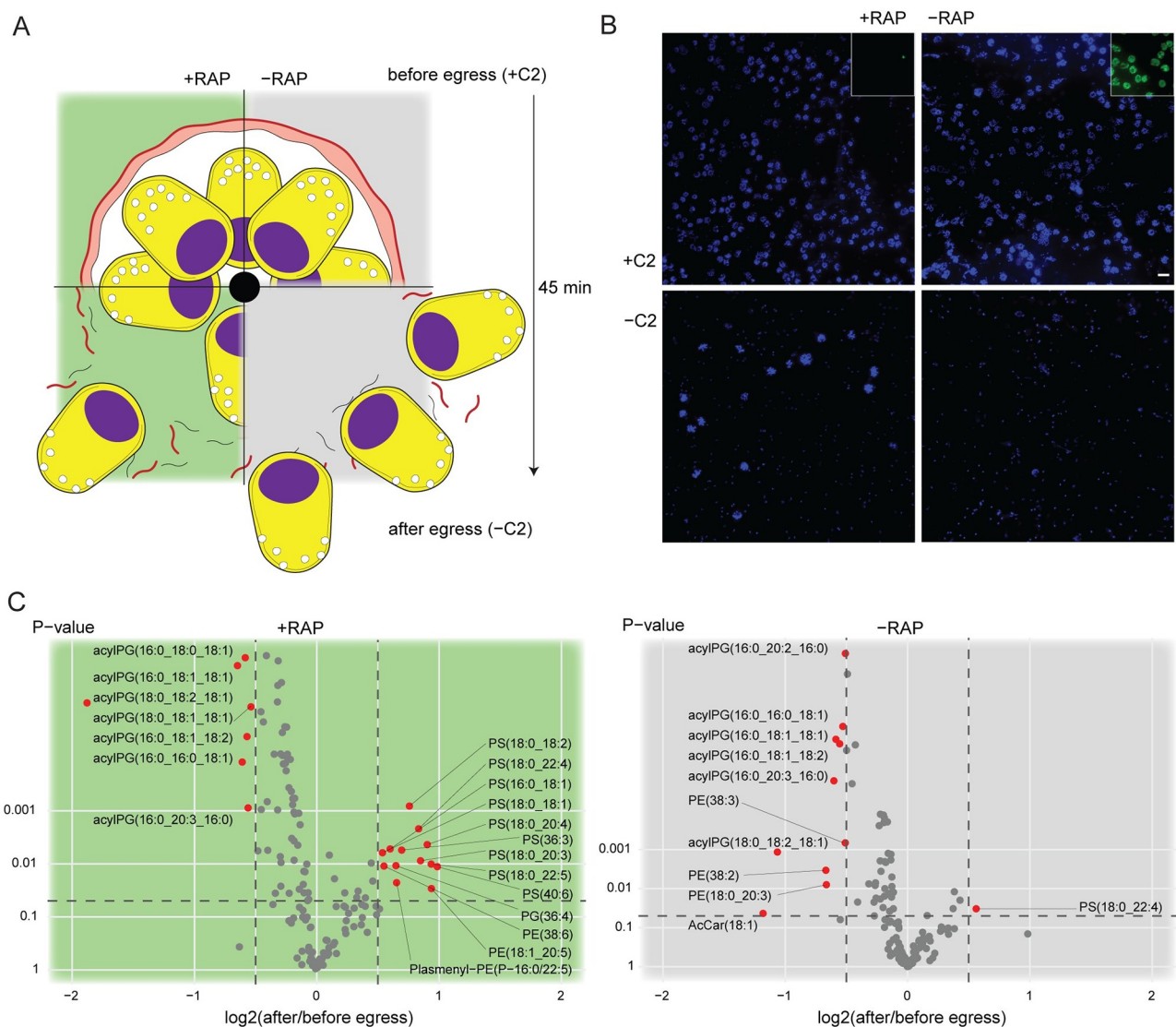

**Fig 6. Lipidomic profiling of LCAT-null parasites reveals changes in phosphatidylserine and acylphosphatidylglycerol levels upon egress. A)** Phospholipid content of RAP-treated (+RAP) and mock-treated (-RAP) LCAT:2loxPint parasites were assessed in egress-stalled schizonts (+C2) and after washing away C2, allowing egress to ensue for 45 min (-C2). -C2 suspensions contained free merozoites, remnants of ruptured PVM (black) and RBCM (red), a few un-egressed schizonts and in addition to this, merozoite clumps in the +RAP cultures due to inefficient egress. **B)** IFA confirming status of egress in +C2 and -C2 samples. The images show intact schizonts in +C2 samples and egressed merozoites (and merozoite clumps in the case of LCAT-null parasites) in -C2 samples. Inset, successful ablation of LCAT (green) expression in RAP-treated parasites. Nuclei were stained with DAPI (blue). Scale bar, 10 μm. **C)** Volcano plots showing fold changes in levels of various phospholipid species occurring during egress of RAP (+RAP) and mock-treated (-RAP) LCAT:2loxPint schizonts. Levels of acylphosphatidylglycerol (acylPG) decrease upon egress of both +RAP and -RAP parasites. An increase in several phosphatidylserine species is observed upon egress of +RAP parasites (comparisons were done across 6 independent egress experiments).

PPLP1 and PPLP2 have established roles in sporozoite egress from transient vacuoles [21] and gametocyte egress from RBCs [23] respectively. The potential role of these proteins in ABS egress, however, has been a matter of debate. Both proteins were shown to be detectable in *P. falciparum* asexual stages with suggested overlapping roles during blood stage egress [24]. In contrast, in other studies, neither PLP was detected in ABS of rodent malaria parasites and individual null mutants exhibited normal blood stage growth [17,19–23]. However, individual

gene knockouts do not rule out the possibility that the PLPs might play compensatory roles, a notion supported by the finding that small molecule PLP inhibitors have anti-parasite activity and block egress [25]. In this work we tested this hypothesis with a PPLP1/PPLP2 double knockout mutant. The PPLP1/PPLP2-null mutants exhibited normal growth rates, membrane poration and egress, confirming their combined dispensability in ABS growth. Our results contradict the findings of [25] and may suggest that the observed egress block induced by the chemical inhibitors used in their study is due to off-target activity. Collectively, the evidence from all functional studies (including this study) of *Plasmodium* PLPs strongly suggests that these molecules have no important role in ABS parasite egress.

The malaria parasite secretes proteins into the PV through specialized apical organelles to perform functions that are vital for egress and invasion as well as modifying the nascent PV after invasion. It is becoming increasingly evident that subpopulations of these organelles are discharged in a controlled way at specific times to perform specific functions [8,39]. Our highly selective proteomics approach, comparing PV-extracts of SUB1-null and control parasites treated with an inhibitor of organelle discharge (C2), identified known microneme and exo-neme proteins that are secreted into the PV within a short timeframe before egress. These include well-established micronemal proteins that are discharged onto the merozoite apical tip (MTRAP; [40]) or merozoite surface (GAMA, EBA-140 and EBA-181; [41–44]) during egress. Surprisingly no differences between the parasite extracts were observed in the levels of another micronemal *ebl* protein, EBA-175 [45], whereas the bona fide exoneme protease plasmepsin X (PMX) that proteolytically processes SUB1 was enriched in the C2-arrested extracts [33,34]. In addition to this, we also found a putative nuclear export mediator factor (NEMF; PF3D7_1202600) enriched 2.5-fold that could potentially be studied further. This gene is con-tinually expressed in ABS with peak transcription during schizont stages and is considered essential [46]. Our approach is by no means exhaustive, and our workflow may have failed to capture low-abundance proteins. Our data, whilst not revealing new effector molecules, still essentially captured the PV proteome during egress and showed the presence of the known phospholipase LCAT in all our PV protein extracts, leading us to study this molecule further.

The role of LCAT has been previously examined in *Toxoplasma gondii* tachyzoites and *P. berghei* sporozoite and liver stages. LCAT orthologs in these organisms were found to be diversely localised depending on the parasite life stage. TgLCAT localises to the plasma mem-brane in extracellular parasites and is secreted into the PV lumen late in the intracellular repli-cation cycle via dense granule-like organelles [47,48]. Similarly, in *P. berghei*, PbLCAT is expressed on the surface of sporozoites [29] but after hepatocyte invasion mainly localises to the PV and the PVM in addition to vesicular structures within the parasite cytoplasm [30]. Loss of LCAT expression results in delayed egress of *T. gondii* tachyzoites from the host cell [47,48], *P. berghei* sporozoites from oocysts, and *P. berghei* merozoites from liver-stage schiz-onts [30].

Here we have shown that LCAT localisation and its facilitative role during egress observed elsewhere can now be extended to *P. falciparum* ABS schizonts too. We localised LCAT both to punctate signals within the cytoplasm (possibly secretory organelles) as well as the PV lumen. Ablation of LCAT expression results in a reduced replication rate, caused by a defective egress phenotype in which RBCM poration is only mildly affected, RBCM rupture appears to progress normally, but merozoite dispersal is inefficient. We can infer from our observations that the PVM ruptures at least partially since the PV protein SERA6 is required to access the internal face of the RBCM for it to rupture. Partial or complete PVM rupture is also strongly supported by the fact that the egress defect can be overcome by mechanical shear stress, pre-sumably helping to disperse the merozoites. In contrast, shear forces failed to rescue the egress defect in SUB1-null or SERA6-null schizonts where an intact RBCM and/or PVM was present

[14]. Inefficient egress in LCAT-null liver stage schizonts was attributed to a defect in PVM disruption, albeit partial as a proportion of LCAT-null parasites were still able to disrupt the PVM [30]. Our observations here indicate that inefficient egress in LCAT-null ABS schizonts cannot be attributed to a defect in PVM rupture.

Our results also contrast with the lack of phenotype observed in our previous loss-of-function analyses in *P. berghei* and *P. falciparum* blood stages [28,30]. This is perhaps not surprising for *P. berghei* ABS as the in vivo environment (akin to mechanical shaking) likely ensures efficient dispersal of the LCAT-null merozoites. The ~15% decrease in growth rate observed after conventional knockout of *P. falciparum* LCAT (as opposed to the ~66% reduction we observed in our conditional knockout line) could be a result of parasite adaptation to its loss in long-term culture. With the inducible DiCre system, we were able to clearly discern the LCAT-null phenotype by studying the mutants immediately after ablating LCAT expression, including within the same erythrocytic cycle.

*P. berghei* LCAT has been shown to hydrolyse PC to produce lysoPC in vitro (at rates significantly lower than human LCAT) [29] suggesting that LCAT could lyse membranes either by direct hydrolysis of PC in the membrane or by producing lysoPC species which themselves are membranolytic [29,47]. However, our extensive lipidomics analysis of LCAT-null schizonts showed no significant changes in PC or lysoPC levels either before or following egress. The nearly identical phospholipid profiles of LCAT-null and wildtype schizonts prior to egress either suggests that LCAT remains inactive before egress or that LCAT activity results in very small changes in phospholipid levels that cannot be discerned by our approach. By contrast, egress of both LCAT-null and wildtype schizonts was accompanied by distinct changes in phospholipid content. The accumulation of phosphatidylserine during LCAT-null schizont egress is intriguing. Phosphatidylserine is significantly enriched in RBC-derived vesicles (RMVs) that are released from infected RBCs [49]. Release of RMVs peaks during schizogony shortly before egress but does not occur during egress [50]. It is plausible that inefficient egress of LCAT-null schizonts extends the period over which RMVs are released, thereby increasing PS content to a higher degree compared to wildtype schizonts. Unfortunately, our lipidomics efforts failed to provide any further insights into the molecular role of LCAT in facilitating efficient egress.

The consistent depletion of acyl-phosphatidylglycerol (acyl-PG) over the course of ABS egress (irrespective of the presence or absence of LCAT) is intriguing. Acyl-PG (also known as semilysobisphosphatidic acid, its stereoisomeric counterpart) is an unusual glycerophospholipid with three acyl chains, one ester-bonded to the glycerol head group and the other two to the glycerol 3-phosphate backbone. The detected acyl-PG species possessed either palmitate (16:0) or stearate (18:0) at the head group while the other two acyl chains were combinations of saturated and unsaturated fatty acids. Acyl-PG was initially identified as a major phospholipid class in bacteria [51,52] but has also been found enriched in Golgi membranes in rodents [53] and is thought to play a role in membrane rupture and assembly during vaccinia virus assembly [54,55]. Acyl-PG has been suggested to play a role in vesicle budding and fusion as its small polar head and three acyl chains gives the molecule a conical shape that can induce membrane curvature [27,56]. Phospholipases are known to interconvert cone-shaped PLs like acyl-PG and inverted cone-shaped PLs like lysoPLs to modulate membrane curvature during endocytosis [57]. In *Plasmodium*, acyl-PG levels have been shown previously to peak in late schizonts [49]. Our results lead us to speculate that acyl-PG could be continuously delivered to the expanding PVM to maintain its curvature, then during egress a phospholipase degrades acyl-PG thereby causing the PVM to rupture. An alternative or additional hypothesis is that acyl-PG could be a major constituent of endocytic vesicles that deliver proteolytic enzymes to the PV prior to egress and their fusion to the PPM or PVM is facilitated by removal of acyl-PG

by a phospholipase. We were unable to pursue these hypotheses further, in part because antibodies against acyl-PG suitable for localisation studies are unavailable and the molecular players in acyl-PG metabolism are largely unknown. Recently, the phospholipase PfPATPL1 was found to play a role in gametocyte egress, with PfPATPL1-null parasites showing defects in rounding up and in vesicular transport of proteins to the parasite periphery [58]. While it is not known whether acyl-PG is deacylated by phospholipases, it is tempting to speculate that this phospholipid species plays an important role during egress.

How the PVM is ruptured during egress is a key question towards understanding the molecular mechanisms that bring about ABS egress. Whilst our diverse efforts here failed to answer this, we have eliminated PPLPs as prospective effectors, established a previously unknown role for the phospholipase LCAT and identified acyl-PG as a phospholipid that undergoes substantial depletion during asexual blood stage egress.

## Methods

### Plasmid construction

Modification plasmids to produce the five modified *P. falciparum* lines used in this study were constructed as follows.

The conditional knockout lines were produced by modifying the endogenous target loci in the DiCre-expressing *P. falciparum* B11 line using Cas9-mediated genome editing [59]. A two-plasmid system was used where a targeting plasmid delivers Cas9 and guide RNA to target loci while a repair plasmid delivers the repair template for homology-directed repair of the Cas9--nicked locus.

The conditional double knockout PPLP1:loxNint/PPLP2:loxPint line was produced by sequentially floxing the endogenous *pplp1* (PF3D7_0408700) and *pplp2* (PF3D7_1216700) loci. Two RNA targeting sequences (TTTTAAAGCATTCTTAAATT for PPLP1 and TTTTTCTAGATATTCACCAA for PPLP2) were inserted into the pDC2 Cas9/gRNA/hDHFR (human dihydrofolate reductase)/yFCU (yeast cytosine deaminase/uridyl phosphoribosyl transferase)- containing plasmid as described previously [60] to generate two different targeting plasmids (pCas9_pplp1_gRNA01 and pCas9_pplp2_gRNA01 respectively). For the repair plasmid pREP-PPLP1 for eight-exon *pplp1*, a recodonised segment of the coding region between second and sixth intron gene (628–2,577 bp; 99–587 aa) flanked by loxN-containing *sera2* introns [61] and ~400–500 bp homology arms was synthesized commercially (GeneArt, Thermo Fisher Scientific) as a 2,361 bp long synthetic DNA fragment. Similarly, for pREP-PPLP2, the recodonised version of the *pplp2*'s MACPF domain (1,127–2,640 bp; 375–812 aa) flanked by loxPint modules [61] and ~500 bp homology arms was synthesized commercially as a 2,497 bp long synthetic DNA fragment. Two different pairs of *lox* sites, the canonical loxP (core sequence- GCATACAT) and its variant loxN (core sequence—GTATACCT), were used for floxing to prevent cross-recombination events between both loci [62]. The repair plasmids were linearised overnight with PvuI and SacI prior to transfection.

The conditional frameshift-based knockout line LCAT:2loxPint, was produced by floxing a 200 bp region (973-1172bp) upstream of the 1,053 bp long catalytic domain that encodes an alpha/beta hydrolase fold with two catalytic GXSXG lipase motifs and a $HX_4D$ acyltransferase motif in the *lcat* gene (PF3D7_0629300). Two RNA targeting sequences (TAATAATAGAGATGAAATTT and ATAGAGATGAAATTTTGGTA) were inserted into the pDC2 Cas9/gRNA/hDHFR-containing plasmid to generate two different targeting plasmids (pCas9_lcat_gRNA01 and pCas9_lcat_gRNA02 respectively). For the repair plasmid pREP-LCAT, a 1,354 bp long synthetic DNA fragment containing a recodonised segment of the 200 bp

flanked by 177 bp long loxPint modules and 400 bp homology arms was synthesized commercially. The repair plasmid was linearised with SphI overnight prior to transfection.

The LCAT:GFP and LCAT:smMyc lines were made by tagging the endogenous *P. falciparum lcat* gene with GFP or smMyc using the selection-linked integration (SLI) method [36]. A GFP-tagging construct pSLI-PF3D7_0629300-GFP was generated by amplifying the C-terminal 900 bp of the *lcat* gene (PF3D7_0629300) using primers PF3D7_0629300-TAG-fw/PF3D7_0629300-TAG-rev and cloning into pSLI-TGD [36] using NotI/MluI.

Similarly, a smMyc-tagging construct pSLI-PF3D7_0629300-SM-Myc was generated by amplifying the smMyc sequence from pCAG-smFP Myc (Addgene plasmid #59757, gift from Loren Looger) [37] using primer smMyc-fw/smMyc-rev and cloned into pSLI-PF3D7_0629300-GFP using MluI/SalI thereby replacing the GFP coding sequence with smMyc.

CloneAmp HiFi PCR Premix (TakaraBio) and Phusion High-Fidelity DNA polymerase (New England BioLabs) were used for PCR reactions for all plasmid constructions. All plasmid sequences were confirmed by Sanger sequencing. For sequences of oligonucleotides and other synthetic DNA species used in this study, please refer to S1 Table.

## Parasite culture maintenance, synchronisation and transfection

The DiCre-expressing *P. falciparum* B11 line [63] was maintained at 37°C in human RBCs in RPMI 1640 containing Albumax II (Thermo Fisher Scientific) supplemented with 2 mM L-glutamine. Synchronisation of parasite cultures were done as described previously [64] by isolating mature schizonts by centrifugation over 70% (v/v) isotonic Percoll (GE Healthcare, Life Sciences) cushions, letting them rupture and invade fresh erythrocytes for 2 h at 100rpm, followed by removal of residual schizonts by another Percoll separation and sorbitol treatment to finally obtain a highly synchronised preparation of newly invaded ring-stage parasites. To obtain the GDPD:HA:loxPint line, transfections were performed by introducing DNA into $\sim 10^8$ Percoll-enriched schizonts by electroporation using an Amaxa 4D Nucleofector X (Lonza), using program FP158 as previously described [65]. For Cas9-based genetic modifications, 20 µg of targeting plasmid and 60 µg of linearised repair template were electroporated. Drug selection with 2.5 nM WR99210 was applied 24 h post-transfection for 4 days with successfully transfected parasites arising at 14–16 days. For sequential genetic modification, the PPLP1:loxPint line was treated with 1 µM 5-fluorocytosine (5-FC) provided as clinical grade Ancotil (MEDA) for one week before transfection with pCas9_pplp2_gRNA01 + pREP-PPLP2.

Clonal transgenic lines were obtained by serial limiting dilution in flat-bottomed 96-well plates [38] followed by propagating wells that contain single plaques. Successful integration was confirmed by running diagnostic PCR either directly on culture using BloodDirect Phusion PCR premix or from extracted genomic DNA (DNAeasy Blood and Tissue kit, Qiagen) with CloneAmp HiFi PCR Premix (TakaraBio).

The *P. falciparum* 3D7 line was maintained at 37°C in an atmosphere of 1% $O_2$, 5% $CO_2$, and 94% $N_2$ and cultured using RPMI complete medium containing 0.5% Albumax according to standard procedures [66]. For generation of stable integrant cell lines LCAT:GFP and LCAT:smMyc, mature 3D7 schizonts were electroporated with 50 µg of plasmid DNA using a Lonza Nucleofector II device [65] and selected in medium supplemented with 3 nM WR99210. WR99210-resistant parasites were subsequently treated with 400 µg/mL Neomycin/G418 (Sigma) to select for integrants carrying the desired genomic modification as described previously [36]. Successful integration was confirmed by diagnostic PCR using FIREpol DNA polymerase (Solis BioDyne).

To obtain LCAT-null parasites, DiCre-mediated excision of the target locus was induced by rapamycin treatment (100 nM RAP for 3 h or 10 nM overnight) of synchronous early ring-stage parasites (2–3 h post-invasion) as previously described [67]. Mock treated parasites were used as wild type controls.

## Western blot and immunofluorescence assays

To detect LCAT protein expression in LCAT:2loxPint line, rabbit polyclonal antibodies were raised against two peptide sequences within the N-terminal region of the *P. falciparum* LCAT protein; SIFLRNPYKITLGKSEK (32–48 aa) and FSEEEDSIVRRDTEKK (56–71 aa). For western blotting, proteins were extracted from C2-arrested mature schizonts directly into SDS buffer. For saponin fractionation, C2-arrested schizonts were treated with 0.05% (w/v) saponin solution in PBS at 37°C for 5 min. Saponin lysates were spun down at maximum speed and proteins were extracted from the resulting supernatant and parasite pellet fractions with SDS buffer. Denatured protein extracts were resolved by SDS polyacrylamide gel electrophoresis (SDS-PAGE) and transferred to nitrocellulose membrane (Supported nitrocellulose membrane, Bio-Rad). Membranes were blocked with 5% bovine serum albumin (BSA) in PBS-T (0.05% Tween 20) and subsequently probed with the rabbit anti-LCAT sera (1:1,000 dilution), followed by horseradish peroxidase-conjugated goat anti-rabbit antibody (BioRad, 1:3,000). Immobilon Western Chemiluminescent HRP Substrate (Millipore) was used according to the manufacturer's instructions, and blots were visualized and documented using a ChemiDoc Imager (Bio-Rad) with Image Lab software (Bio-Rad). Rabbit antibodies against SERA6 [11] was used at 1:1,000 as loading control.

For immunofluorescence assays of LCAT:2loxPint parasites, thin films of parasite cultures containing C2-arrested mature schizonts were air-dried, fixed in 4% (w/v) formaldehyde for 30 min (Agar Scientific Ltd.), permeabilized for 10 minutes in 0.1% (w/v) Triton X-100 and blocked overnight in 3% (w/v) bovine serum albumin in PBS. Slides were probed with rabbit anti-LCAT sera (1:1,000 dilution), followed by AlexaFluor 488-conjugated anti-rabbit antibodies (Invitrogen, 1:1,000). Slides were then stained with 1 μg/mL DAPI and mounted in Citifluor (Citifluor Ltd., Canterbury, U.K.).

For imaging LCAT:smMyc parasites, a circular space on a coverslip was located with a Dako pen and coated with 10 μl 0.5 mg/ml Concanavalin A (ConA, in $H_2O$, Sigma) in a humid chamber for 15 min at 37°C. 500 μl of Compound 2 (C2, 1 μM) arrested LCAT: smMyc schizont cultures were centrifuged and washed once in PBS to remove media components. ConA was washed away (3x $H_2O$, 3x PBS) from coverslips and 50 μl of parasite cultures were applied to the coverslips and incubated at 37°C for 15 min in a humid chamber. Unbound cells were washed away with PBS and bound cells were fixed with 2% PFA / 0.0065% glutaraldehyde in PBS for 20 min at RT. Until this point, all solutions contained 1 μM C2 to prevent schizont egress. Cells were washed 3x with PBS and permeabilized with 0.2% Triton X-100 in PBS for 10 min at RT. Cells were then blocked for 10 min with 3% BSA/PBS followed by an incubation with 30 μl of primary antibody rabbit anti-Myc (1:1.000, Cell Signalling Technology, #2272) in 3% BSA/PBS in a humid chamber for 1 h at RT. After washing cells 3x with PBS, cells were incubated in a humid chamber for 1 h at RT with 30 μl of secondary antibody donkey anti-rabbit AlexaFluor488 (1:1.000, Invitrogen) in 3% BSA/ PBS additionally containing 1 μg/ml DAPI for visualization of nuclei. After washing cells 3x with PBS, slides were mounted with Dako mounting solution, sealed and stored at 4°C protected from light until analysis.

For colocalisation analysis of LCAT:GFP with markers for micronemes and rhoptries, air dried thin blood films were fixed for 3 min in ice-cold methanol and rehydrated in PBS. For

colocalisation analysis of LCAT:GFP and the exoneme marker SUB1, C2-arrested LCAT-GFP schizonts were fixed in 4% PFA in PBS, washed twice in PBS, permeabilized for 15 min at room temperature using 0.5% Triton X-100 in PBS and washed another two times in PBS. Blocking of all samples was performed in 3% BSA/PBS, followed by staining in blocking buffer with the following primary antibodies: rabbit anti-GFP (ChromoTek, 1:1,000); mouse anti-AMA1 antibody 1F9 [68] (1:1,000); mouse monoclonal anti-RAP1 antibody 2.29 [69](1:1,000); or mouse monoclonal anti-SUB1 antibody NIMP.M7 [7] (1:1,000). After washing with PBS, slides were probed with the following secondary antibodies in blocking buffer additionally containing 1 μg/ml DAPI: goat anti-rabbit-AlexaFluor488 antibody (Invitrogen, 1:1,000) or goat anti-mouse-AlexaFluor594 antibody (Invitrogen, 1:1,000). After a final washing step with PBS, DAKO mounting solution was added and slides were covered with a coverslip for imaging.

## Fluorescence and time-lapse microscopy

For live cell microscopy of LCAT:GFP parasites, parasites were incubated with 1 μg/mL DAPI in culture medium for 15 min at 37˚C to stain nuclei before microscopic analysis. Parasites were imaged on a Leica D6B fluorescence microscope, equipped with a Leica DFC9000 GT camera and a Leica Plan Apochromat 100x/1.4 oil objective. Viewing chambers for live microscopy were constructed by adhering 22 × 64 mm borosilicate glass coverslips (VWR International) to microscope slides, as described previously [67]. Mature Percoll-enriched schizonts were incubated for $\geq$ 4 h at 37˚C in Albumax-supplemented RPMI medium supplemented with 1 μM C2. Subsequently, ~$5 \times 10^7$ schizonts were rapidly washed twice in 1 ml of gassed complete medium pre-warmed to 37˚C and lacking C2, pelleting at $1,800 \times g$ for 1 min. The cells were suspended in 60 μl of the same medium and introduced into a pre-warmed viewing chamber which was then immediately placed on a temperature-controlled microscope stage held at 37˚C on a Nikon Eclipse Ni-E wide-field microscope fitted with a Nikon N Plan Apo λ 100×/1.45NA oil immersion objective and a Hamamatsu C11440 digital camera and documented via the NIS Elements software (Nikon). Images were acquired at 5 to 10 s intervals over a total of 30–40 min then processed and exported as TIFFs using Fiji [70]. For simultaneous capture of egress of RAP- and mock-treated schizonts in the same chamber, one of the lines were stained with 1μg/μL Hoechst in C2-supplemented media at 37˚C for 5 min and washed twice with 1 mL C2-supplemented media before adding to an equal amount of unstained line and proceeding as before. RAP- and mock-treated lines were stained alternatively for each video. For visualising RBCM poration, schizonts were washed and then suspended in 60 μL media with E64 (50 μM final concentration) and AlexaFluor 488 phalloidin (Invitrogen; diluted 1:50 from 200 unit ml$^{-1}$ stock in PBS) and proceeded as before for imaging. For flow cytometry-based quantification of porated RBCs, egress-arrested schizonts were stained with Hoechst in C2-supplemented media at 37˚C for 5 min and washed once with 1 mL C2-supplemented media. Hoechst-stained schizonts were washed and then resuspended in either 200 μL media with E64 and AlexaFluor 488 phalloidin and allowed to egress for 30 min at 37˚C in tubes. As a negative control, another aliquot of the schizonts were incubated in the presence of C2 instead of E64. Samples before and after incubation (t = 0 and t = 30min) were analysed by flow cytometry on a BD FACSVerse using BD FACSuite software. For every sample, 10,000–30,000 events were recorded and filtered with appropriate forward and side scatter parameters. Hoechst-positive (infected RBCs) were gated using a 448/45 detector configuration and of this, AlexaFluor 488 phalloidin-positive RBCs were counted using a 527/32 detector configuration. All data were analysed using FlowJo software.

## Growth and replication assays

Growth assays were performed to assess parasite growth across 3–4 erythrocytic replication cycles. Synchronous cultures of ring-stage parasites at 0.1% parasitaemia and 2% haematocrit were maintained in triplicates in 12 well plates. 50 μL from each well was sampled at 0, 2, 4 and 6 days post-RAP treatment, fixed with 50 μL of 0.2% glutaraldehyde in PBS and stored at 4˚C for flow cytometry quantification. Fixed parasites were stained with SYBR Green (Thermo Fisher Scientific, 1:10,000 dilution) for 20 min at 37˚C and analysed by flow cytometry on a BD FACSVerse using BD FACSuite software. For every sample, parasitaemia was estimated by recording 10,000 events and filtering with appropriate forward and side scatter parameters and gating for SYBR Green stain-positive (infected RBCs) and negative RBCs using a 527/32 detector configuration. All data were analysed using FlowJo software. Growth stage progression was monitored by microscopic examination at selected timepoints using Giemsa-stained thin blood films.

Plaque growth assays were performed by dispensing around 20 parasites per well in flat-bottomed microplates at a haematocrit of 0.75%, as described [38]. Plates were imaged using a high resolution flat-bed scanner 14–16 days after setting up the assays. Plaques were counted by visual examination of the images and plaque size quantified using the Lasso tool in Adobe Photoshop CS6.

To assess invasion rates, highly synchronous mature schizonts were added to fresh erythrocytes (2% haematocrit) and allowed to invade for 4 h under either static conditions or with mechanical shaking (100 rpm) (four replicates in each condition). Cultures were sampled before and after the 4 h invasion period and fixed as before for quantification.

## Proteomic analysis

To assess the changes in PV proteome upon micronemal/exonemal release, synchronous SUB1HA3:loxP parasites [14] at ~32 hpi were treated with 1 μM C2 overnight. To trigger micronemal/exonemal release, schizonts were washed with RPMI without Albumax and incubated in RPMI without Albumax for 20 min at 37˚C. C2-treated and C2-washed schizonts were lysed in ice cold 0.15% (w/v) saponin (with the addition of C2 for the +C2 samples). The saponin fractions were filtered using Ultrafree-MC 0.22 μm GV Durapore (Milipore) filters centrifuged at 13,000 rpm for 1 min. Multiple aliquots of the saponin fractions were snap frozen on dry ice/ethanol. Proteins were denatured by adding 20 μl of 4x Laemmli buffer with 10 mM DTT freshly added and heating at 95˚C for 5 min. Denatured proteins were run on a BioRad TGX 4–15% Tris-Glycine gel and then proceeded with in-gel digestion.

Reduced and alkylated proteins were in-gel digested with 100 ng trypsin (modified sequencing grade, Promega) overnight at 37˚C. Supernatants were dried in a vacuum centrifuge and resuspended in 0.1% trifluoroacetic acid (TFA).

On an Ultimate 3000 nanoRSLC HPLC (Thermo Scientific) 1–10 μl of acidified protein digest was loaded onto a 20 mm x 75 μm Pepmap C18 trap column (Thermo Scientific) prior to elution via a 50 cm x 75 μm EasySpray C18 column into a Lumos Tribrid Orbitrap mass spectrometer (Thermo Scientific). A 90 min binary gradient of 6%-40% B over 63 min was used prior to washing and re-equilibration (buffer A, 2% ACN, 0.1% formic acid; buffer B, 80% ACN, 0.1% formic acid).

The Orbitrap was operated in 'TopS' Data Dependent Acquisition mode with precursor ion spectra acquired at 120k resolution in the Orbitrap detector and MS/MS spectra at 32% HCD collision energy in in the ion trap. Automatic Gain Control was set to Auto for MS1 and MS2. Maximum injection times were set to 'Standard' (MS1) and 'Dynamic' (MS2). Dynamic exclusion was set to 20 s.

Raw files were processed using Maxquant (maxquant.org) and Perseus (maxquant.net/perseus) with recent downloads of the Plasmodium falciparum 3D7 (www.plasmodb.org) and the Uniprot Homo sapiens reference proteome, together with the Maxquant common contaminants databases. A decoy database of reversed sequences was used to filter false positives at protein and peptide FDR of 1%. T-tests were performed with a permutation-based FDR of 5% to cater for multiple hypothesis testing.

## Lipidomic analysis

To assess the changes in phospholipid content due to absence of LCAT, total phospholipids from LCAT-null and wildtype schizonts were extracted and lipid species were determined and quantified by LC-MS/MS. Schizonts were isolated using Percoll cushions from RAP- and mock-treated LCAT:2loxPint parasitised cultures (100 ml, 0.5% haematocrit, 35–40% parasitaemia) grown for 45 h post treatment and allowed to mature for 4 h at 37˚C in the presence of C2 (1 μM) in order to achieve a high level of homogeneity in the samples. Arrested schizonts were washed twice with RPMI media without Albumax II (including C2 at 1 μM) and subject to lipid extraction. Lipid extraction for each sample was performed by adding 400 μL of $1 \times 10^8$ parasites, either as intact schizonts or egressed suspensions, to each of three tubes (technical replicates) that contained 600 μL methanol and 200 μL chloroform. Experiments were carried out in triplicate. To assess the changes in phospholipid content upon egress, egress-blocked LCAT-null (+RAP) and wildtype (-RAP) schizonts (in RPMI media without Albumax II with C2 at 1 μM) were divided into 12 aliquots and kept at 37˚C. Of these, 6 aliquots were washed with prewarmed RPMI media w/o Albumax and incubated in 200 μL prewarmed RPMI media w/o Albumax at 37˚C for 45 min for them to egress. Total phospholipids were extracted from the six egress-blocked samples (+C2) and from the six independently egressed samples (-C2) by adding 200 μL of approximately $1 \times 10^{10}$ parasites, either as intact schizonts or egressed suspensions, to 600 μL methanol and 200 μL chloroform. Samples were sonicated for 8 min at 4˚C and incubated at 4˚C for 1 h. 400 μL of ice-cold water was added (thus obtaining the 3:3:1 water:methanol:chloroform ratio) to the samples, mixed well and centrifuged at max speed for 5 min at 4˚C for biphasic partitioning. The lower apolar phase was added to fresh tubes. The upper aqueous layer was removed and lipids were extracted once more by adding 200 μL of chloroform, vortexing and centrifuging as before. The apolar phases from both extractions were pooled (400 μL) and dried under nitrogen stream and resuspended in butanol/methanol (1:1,v/v) containing 5 μM ammonium formate.

The LC-MS method was adapted from [71]. Cellular lipids were separated by injecting 10 μL aliquots onto a column: $2.1 \times 100$ mm, 1.8 μm C18 Zorbax Elipse plus column (Agilent) using an Dionex UltiMate 3000 LC system (Thermo Scientific). A 20 min elution gradient of 45% to 100% solvent B was used, followed by a 5 min wash of 100% solvent B and 3 min re-equilibration, where solvent B was water:acetonitrile:isopropanol, 5:20:75 (v/v/v) containing 10 mM ammonium formate (Optima HPLC grade, Fisher Chemical) and solvent A was 10 mM ammonium formate in water (Optima HPLC grade, Fisher Chemical). Other parameters were as follows: flow rate 600 μL/min; column temperature 60˚C; autosampler temperature 10˚C. MS was performed with positive/negative polarity switching using an Q Exactive Orbitrap (Thermo Scientific) with a HESI II probe. MS parameters were as follows: spray voltage 3.5 kV and 2.5 kV for positive and negative modes, respectively; probe temperature 275˚C; sheath and auxiliary gases were 55 and 15 arbitrary units, respectively; full scan range: 150 to 2000 m/z with settings of auto gain control (AGC) target and resolution as Balanced and High ($3 \times 10^6$ and 70,000), respectively. Data was recorded using Xcalibur 3.0.63 software (Thermo Scientific). Mass calibration was performed for both ESI polarities before analysis using the standard Thermo

Scientific Calmix solution. To enhance calibration stability, lock-mass correction was also applied to each analytical run using ubiquitous low-mass contaminants. To confirm the identification of significant features, pooled quality control samples were run in data-dependent top-N (ddMS2-topN) mode, acquisition parameters as follows: resolution of 17,500, auto gain control target under $2 \times 10^5$, isolation window of m/z 0.4 and stepped collision energy 10, 20 and 30 in HCD (high-energy collisional dissociation) mode. Data analysis was performed using Free Style 1.5 (ThermoScientific), Progenesis (Nonlinear Dynamics) and LipidMatch [72].

## Statistical analysis

All statistical analysis and data visualization was performed in R v4.0.2 (R Core Team (2021)). Unless stated otherwise, Student's t-test were used to compare group means and where necessary Bonferroni adjustment for multiple comparisons was applied to the p-value of statistical significance. All statistical analysis is available as R code in https://github.com/a2g1n/LCATxcute.

## Supporting information

**S1 Fig. A)** Diagnostic PCR showing correct integration of the modification plasmids into the PPLP1 and PPLP2 loci in PPLP1:loxNint/PPLP2:loxPint parasites. Primers used are denoted in Fig 1A. **B)** Replication of PLP1:loxPint clonal line prior to second modification. The modified parasites show a normal replication rate across two cycles (error bars, ± SD, triplicate RAP treatments with different blood sources).
(TIF)

**S2 Fig. A)** Strategy for SLI-based endogenous tagging of the *lcat* gene with GFP. Primers used for integration PCR are indicated with half arrows. T2A, skip peptide; Neo-R, neomycin-resistance gene; hDHFR, human dihydrofolate reductase; lollipop, stop codons; arrows, promoters. **B)** Diagnostic PCR showing correct integration of the modification plasmid into the LCAT locus in the LCAT:GFP parasites. KI, knock in cell line; WT, wild type parental line. **C)** Strategy for SLI-based endogenous tagging of *lcat* gene with smMyc. Primers used for integration PCR are indicated with half arrows. **D)** Diagnostic PCR showing correct integration of the modification plasmid into the LCAT locus in the LCAT:smMyc line. **E)** IFA of LCAT:smMyc mature schizonts using anti-myc (green) antibodies showing similar localisation of LCAT as observed in LCAT:GFP line (Fig 3). DAPI-stained nuclei are shown in blue. DIC, differential interference contrast. Scale bar, 5 μm. F) Colocalisation analysis of LCAT:GFP (green) with markers for rhoptries (RAP1), micronemes (AMA1) and exonemes (SUB1) in C2-arrested schizonts. DAPI-stained nuclei are shown in blue. DIC, differential interference contrast. Scale bar, 5 μm. G) Western blot using anti-LCAT polyclonal antibodies shows distribution of LCAT upon saponin-mediated fractionation (using treatment with 0.05% saponin at 37˚C for 5 min) of schizonts. LCAT was predominantly detected in the supernatant fraction (+SAP "S") compared to the pellet fraction (+SAP "P"), similar to the soluble PV marker SERA6. The appropriate band size of LCAT is shown from whole schizonts (WS) for comparison (same lane also shown as -RAP in Fig 4C).
(TIF)

**S3 Fig. A)** Diagnostic PCR showing correct integration of the modification plasmid into the LCAT locus in LCAT:2loxPint line. Primers used are indicated in Fig 4A.
(TIF)

**S4 Fig. A)** Lipidomic analysis of LCAT:2loxPint egress-stalled schizonts following mock-or RAP-treatment at ring stages. The bubble plot shows the fold change (y-axis) in levels of

various lipid species (each lipid class denoted with a different colour) in LCAT-null schizonts compared to controls (3 independent biological replicates). No significant change in phospholipid levels were detected between the samples. **B)** Bubble plot showing the fold change in levels of various lipid species before and after egress of RAP-treated (+RAP) and mock-treated (-RAP) LCAT:2loxPint parasites (6 independent biological replicates).
(TIF)

**S5 Fig.** Relative peak intensities (depicted as barplots) and $log_2$ fold change (depicted as dot plots) of the significantly altered **A)** phosphatidylserine and **B)** acylphosphatidylglycerol species upon egress of mock- or RAP-treated LCAT:2loxPint schizonts.
(TIF)

**S1 Table. Sequences of oligonucleotides and other synthetic DNA used in this study.**
(XLSX)

**S2 Table. Label-free quantitation of proteins detected in SUB1-null schizonts in the presence or absence of C2.** T-test analysis comparing +C2 and -C2 samples. Normalised spectral peak intensities in each replicate are provided.
(XLSX)

**S3 Table. Raw peak intensities of various lipid species measured in LCAT:2loxPint egress-stalled schizonts following mock-or RAP-treatment at ring stages.** T-test analysis comparing +RAP and -RAP schizonts and normalised spectral peak intensities in each replicate are provided.
(XLSX)

**S4 Table. Raw peak intensities of various lipid species measured before and after egress in RAP-treated (+RAP) and mock-treated (-RAP) LCAT:2loxPint parasites.** T-test analyses comparing the different groups and normalised spectral peak intensities in each replicate are provided.
(XLSX)

**S1 Movie. Composite time-lapse video showing RAP- and mock-treated (blue; stained with DAPI) PPLP1:loxNint/PPLP2:loxPint parasites undergoing normal RBCM poration and egress.** Poration of the RBCM is visualized as an accumulation of a fluorescent phalloidin (green) signal around the RBC circumference just prior to egress. Scale bar, 5 μm.
(MP4)

**S2 Movie. Composite time-lapse video showing different fates of RAP- (blue; stained with Hoechst) and mock-treated LCAT:2loxPint parasites.** A great number of abnormal egress events where merozoites remain clumped together after egress (marked with red circles; normal egress events are marked in green) are observed in RAP-treated LCAT:2loxPint parasites compared to mock-treated parasites. Scale bar, 10 μm.
(MP4)

**S3 Movie. Composite time-lapse video showing normal RBCM poration in both RAP- (blue; stained with Hoechst) and mock-treated LCAT:2loxPint parasites.** Poration of the RBCM is visualized as an accumulation of a fluorescent phalloidin (green) signal around the RBC circumference and occurs in both Hoechst stained (RAP+) and unstained (RAP-) schizonts. Scale bar, 10 μm.
(MP4)

## Acknowledgments

We thank Michael Foley for providing the monoclonal AMA1 antibody 1F9. Images were acquired on microscopes of the CSSB imaging facility.

## Author Contributions

**Conceptualization:** Abhinay Ramaprasad, Paul-Christian Burda, Konstantinos Koussis, Tim-Wolf Gilberger, Michael J. Blackman.

**Data curation:** Abhinay Ramaprasad, Paul-Christian Burda, Enrica Calvani, Steven A. Howell.

**Formal analysis:** Abhinay Ramaprasad, Paul-Christian Burda, Enrica Calvani, Steven A. Howell.

**Funding acquisition:** Tim-Wolf Gilberger, Michael J. Blackman.

**Investigation:** Abhinay Ramaprasad, Paul-Christian Burda, Konstantinos Koussis, James A. Thomas, Emma Pietsch, Enrica Calvani, Steven A. Howell, James I. MacRae, Ambrosius P. Snijders, Tim-Wolf Gilberger, Michael J. Blackman.

**Methodology:** Abhinay Ramaprasad, Paul-Christian Burda, Konstantinos Koussis, James A. Thomas, Emma Pietsch, Enrica Calvani, Steven A. Howell, James I. MacRae, Ambrosius P. Snijders, Tim-Wolf Gilberger, Michael J. Blackman.

**Supervision:** Tim-Wolf Gilberger, Michael J. Blackman.

**Visualization:** Abhinay Ramaprasad.

**Writing – original draft:** Abhinay Ramaprasad.

**Writing – review & editing:** Abhinay Ramaprasad, Paul-Christian Burda, Konstantinos Koussis, Tim-Wolf Gilberger, Michael J. Blackman.

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
