## [Decision Letter · Decision Letter 0]

1 May 2023

Dear Dr Ramaprasad,

Thank you very much for submitting your manuscript "A malaria parasite phospholipase facilitates efficient asexual blood stage egress" for consideration at PLOS Pathogens. As with all papers reviewed by the journal, your manuscript was reviewed by members of the editorial board and by several independent reviewers. In light of the reviews (below this email), we would like to invite the resubmission of a significantly-revised version that takes into account the reviewers' comments.

We cannot make any decision about publication until we have seen the revised manuscript and your response to the reviewers' comments. Your revised manuscript is also likely to be sent to reviewers for further evaluation.

Sincerely,

Bjorn F.C. Kafsack, Ph.D.

Section Editor

PLOS Pathogens

James Collins III

Section Editor

PLOS Pathogens

Kasturi Haldar

Editor-in-Chief

PLOS Pathogens

orcid.org/0000-0001-5065-158X

Michael Malim

Editor-in-Chief

PLOS Pathogens

orcid.org/0000-0002-7699-2064

Reviewer's Responses to Questions

**Part I - Summary**

Reviewer #1: The manuscript by Ramaprasad and colleagues investigates the role two perforin-like proteins in Plasmodium falciparum egress. After demonstrating that PLP1 and PLP2 are not required, the group more deeply investigates the role of a lecithin:cholesterol acyltransferase protein (LCAT). LCAT knockout led to reduced parasite growth with resulting changes in lipid profile of parasites.

The use of two different Cre recombinase sites, loxP and loxN, is a nice way to delete the two PLP genes at the same time without interference. As noted, this was not necessary as they inadvertently frameshifted PLP1 during the first transfection. The dual deletion parasite strain demonstrates that neither PLP1 or 2 are needed for asexual replication.

The proteomic approach to identify microneme/exoneme proteins uses SUB1-null parasites in the presence and absence of C2. This was the starting point to investigate LCAT in asexual parasites. The phenotyping of the LCAT mutant is interesting and suggests a difference in the lipid composition. However, no follow up studies are performed to verify this finding. Therefore, the reader is left without knowing what the impact of the increased PS and PE species in LCAT-null parasites means for parasite egress.

The findings of the study are different from the initial studies of LCAT in P. berghei where Bhanot, et al. reported no major growth defects in asexual parasites. However, the finding of delayed egress and impaired PVM rupture is the same phenotype as described in Pb liver stage parasites by Burda in 2015. In this second paper, a major defect in asexual growth was also not described.

Reviewer #2: Egress from infected red blood cells is a critical step in the lifecycle of malaria parasites. Although a variety of parasite egress factors have been identified for this event, the known factors cannot account for all the observed events that accompany egress. Previous reports implied that perforin-like proteins 1 and 2 might be expressed in late stage schizonts and contribute to egress in a manner shown for other stages of the parasite and related parasites. The current study advances knowledge of this topic by: (1) definitively showing that PfPLP1 and PfPLP2 are not necessary for normal egress; (2) developing an innovative proteomics screen that identified secretory proteins that are released into the supernatant during egress, including a phospholipase, LCAT, that has been partially characterized previously; (3) showing that conditional KO of LCAT markedly reduces efficient egress, thereby blunting proliferation; and (4) identifying PS and PE as phospholipids that are depleted in an LCAT-dependent manner during egress. By shifting the focus on pore-forming proteins to a phospholipase, this study provides a new perspective on how malaria parasites exit from cells to advance the infection. Addressing the following comments should further strengthen the study.

Reviewer #3: In this paper, the authors investigated molecular events involved in asexual blood stage egress of the deadly parasite Plasmodium falciparum.

This topic is really important in the apicomplexa field but also for the parasite community in general.

As a general comment, the paper is well written, experiments are well presented and done rigorously with appropriate controls and humility in their interpretation.

Even if the authors did not solve completely their problematics they pointed out several issues that will help to better understand the topic but also, with no doubts, will serve for future studies.

4 main points are raised:

-Two performing-like proteins are not involved in egress

-Identification of proteins discharged in the parasitophorous vacuole during egress

-Potent role of a phospholipase in the egress

-Role of acyl-phosphatidyl glycerol during egress.

Strenghts: robust and elegant experiments adding more pieces in the understanding of egress

Weakness: The paper is not really focused.... the way the authors explain why their work on a phospholipase is not intuitive..and after in some extent "negative" results.

The title do not reflect properly the content of the paper. I would suggest to modify the title. In my point of view, the focus on the phospholipase is not justified as it is only part of the demo.

**Part II – Major Issues: Key Experiments Required for Acceptance**

Reviewer #1: MAJOR:

1. The localization data shown (fig 3a, 3b, and sup 2) should be improved some. The hypothesis, which is very likely to be correct, is that LCAT is transported through the secretory system and resides in the PV. The immunofluorescence does not show this clearly. The LCAT and SERA5 staining are diffuse and overlap with MSP1. As shown, the staining could also be interpreted as cytoplasmic. The other data strongly support the hypothesis that LCAT is in the PV. The immunofluorescence should be improved to strengthen this hypothesis even more.

2. The egress phenotype is described as abnormal. The meaning of abnormal needs to be stated so that the reader can understand the quantification in 5D. Similarly, for 5B, what percentage of the egress events had joined merozoites? This is presumably what is being quantified in 5D, but this is not stated.

3. Would it be interesting to compare the after egress -RAP to the after egress +RAP to look for differences that are specific to LCAT? The current analysis looks at after/before egress for each condition. Also, for the lipid analysis, it would be helpful to include some additional description in the text of what the enzymatic function of LCAT is, what types of lipids it is expected to modify, etc. For the non-expert, the meaning and importance of the lipid analysis and the changes is tough to appreciate. If the changes are important, this is not fully conveyed in the text.

Reviewer #2: Main comments

1. The authors conclude that LCAT localizes to secretory vesicles. However, the evidence for this is limited to one fluorescent micrograph showing the distribution of LCAT-GFP possibly being within the parasite and a second micrograph indicating possible relocation of LCAT-GFP to the PV. The authors presumably have good antibodies to microneme, rhoptry, and exoneme markers that could better define where LCAT resides within the parasite. Doing so would help broaden understanding the contributions of various secretory organelles to egress.

2. Related to point #1, data from screens is typically validated by performing a similar experiment that more selectively looks at the target of interest. In this manner it would be ideal to redo the C1/saponin experiment and test s/n’s for LCAT (untagged) and LCAT-GFP. Comparing pellets to S/N’s would also allow assessment of potential proteolytic processing of LCAT as a possible means of regulating its activity.

Reviewer #3: (No Response)

**Part III – Minor Issues: Editorial and Data Presentation Modifications**

Reviewer #1: MINOR

1. There are formatting errors for the references. The Burda bioRxiv paper is present twice (2021a and 2021b).

2. The source of the SUB1-null parasites is not discussed. Is this an inducible SUB1 knockout?

3. The identity of the remaining significant red dot proteins should be indicated in figure 2c. Currently there are identifications provided for only a subset of the proteins.

4. Fig 4F seems to provide more information than is valuable for the reader. Although briefly explained in the legend, the meaning of different features of the graph remains unclear. This is strong data showing the replication defect. It should be presented in a more straightforward manner.

5. Page 11 “solely ascribed” is a strong phrase. This should be worded less definitively.

6. Fig 5F, the image looks strange. The phalloidin image does not look homogenous in the field of view. This makes interpretation difficult. This should be repeated, potentially with better alignment or adjustment.

7. In Supp movies, the use of Hoechst staining to differentiate the two conditions is clever. This should be stated somewhere besides the legend for the supplemental movies. I needed to watch the movies, consider what was going on, then re-read the legends again. This is minor, but it would be helpful to mention this in the text as well.

Reviewer #2: Minor comments

Page 4, 2nd line from bottom: citation appears to be incorrect.

Page 7, 3rd paragraph, 2nd line: “…no effect on parasite proliferation…” Consider being more precise here and use the term “egress” instead of “proliferation” if the studies assessed egress.

Page 9, 1st paragraph, 7th line: should singular/plural misalignment. Should be candidate and was or candidate and were.

Fig 2b. Please label the dot corresponding to LCAT. Also, since MSP1 and MSP2 were identified as being significantly enriched, the authors should mention this and disclose that not all of the proteins identified were from apical secretory organelles.

Page 9, 2nd paragraph, 3rd line: “To confirm the localization…” The expression data doesn’t indicate localization. Should be “To investigate the localization…”

Page 19, 1st paragraph, last line: the importance of acyl-PG is not addressed in the study. Rather, the abundance of this acyl-PG decreases during egress. Additional work would be necessary to define its importance.

Reviewer #3: The authors should detail more the caption in the figures and order to facilitate the understanding, in particular in the movies.

The authors should give more details in the analysis of MS data. The link between the data in the su tale S2 and the volcano plot is not obvious. Give more details.

PLOS authors have the option to publish the peer review history of their article (what does this mean?). If published, this will include your full peer review and any attached files.

Reviewer #1: No

Reviewer #2: **Yes: **Vern Carruthers

Reviewer #3: No
---

## [Editor Report · Decision Letter 1]

31 May 2023

Dear Dr Ramaprasad,

We are pleased to inform you that your manuscript 'A malaria parasite phospholipase facilitates efficient asexual blood stage egress' has been provisionally accepted for publication in PLOS Pathogens.

Best regards,

Bjorn F.C. Kafsack, Ph.D.

Section Editor

PLOS Pathogens

James Collins III

Section Editor

PLOS Pathogens

Kasturi Haldar

Editor-in-Chief

PLOS Pathogens

orcid.org/0000-0001-5065-158X

Michael Malim

Editor-in-Chief

PLOS Pathogens

orcid.org/0000-0002-7699-2064
---

## [Editor Report · Acceptance letter]

21 Jun 2023

Dear Dr Ramaprasad,

We are delighted to inform you that your manuscript, "A malaria parasite phospholipase facilitates efficient asexual blood stage egress," has been formally accepted for publication in PLOS Pathogens.

Best regards,

Kasturi Haldar

Editor-in-Chief

PLOS Pathogens

orcid.org/0000-0001-5065-158X

Michael Malim

Editor-in-Chief

PLOS Pathogens

orcid.org/0000-0002-7699-2064